# l-Asparaginase Type II from *Fusarium proliferatum*: Heterologous Expression and In Silico Analysis

**DOI:** 10.3390/pharmaceutics15092352

**Published:** 2023-09-20

**Authors:** Samuel Leite Cardoso, Paula Monteiro Souza, Kelly Rodrigues, Isabella de Souza Mota, Edivaldo Ferreira Filho, Léia Cecilia de Lima Fávaro, Felipe Saldanha-Araujo, Mauricio Homem-de-Mello, Adalberto Pessoa, Dâmaris Silveira, Yris Maria Fonseca-Bazzo, Pérola Oliveira Magalhães

**Affiliations:** 1Health Science School, University of Brasilia, Brasilia 70910-900, Brazil; samuel.leite@aluno.unb.br (S.L.C.); paulasouza@unb.br (P.M.S.); isabellasmota11@gmail.com (I.d.S.M.); felipearaujo@unb.br (F.S.-A.); mauriciohmello@unb.br (M.H.-d.-M.); damaris@unb.br (D.S.); yrisfonseca@hotmail.com (Y.M.F.-B.); 2Brazilian Agricultural Research Corporation—EMBRAPA Agroenergia, Brasilia 70770-901, Brazil; kellybiobarreto@gmail.com (K.R.); leia.favaro@embrapa.br (L.C.d.L.F.); 3Institute of Biological Sciences, University of Brasilia, Brasilia 70910-900, Brazil; eximenes@unb.br; 4Department of Biochemical and Pharmaceutical Technology, University of São Paulo, São Paulo 05508-000, Brazil; pessoajr@usp.br

**Keywords:** l-asparaginase, in silico analysis, heterologous expression, *Fusarium proliferatum*

## Abstract

The search for new drug-producing microorganisms is one of the most promising situations in current world scientific scenarios. The use of molecular biology as well as the cloning of protein and compound genes is already well established as the gold standard method of increasing productivity. Aiming at this increase in productivity, this work aims at the cloning, purification and in silico analysis of l-asparaginase from *Fusarium proliferatum* in Komagataella phaffii (*Pichia pastoris)* protein expression systems. The l-asparaginase gene (NCBI OQ439985) has been cloned into *Pichia pastoris* strains. Enzyme production was analyzed via the quantification of aspartic B-hydroxamate, followed by purification on a DEAE FF ion exchange column. The in silico analysis was proposed based on the combined use of various technological tools. The enzymatic activity found intracellularly was 2.84 IU/g. A purification factor of 1.18 was observed. The in silico analysis revealed the position of five important amino acid residues for enzymatic activity, and likewise, it was possible to predict a monomeric structure with a C-score of 1.59. The production of the enzyme l-asparaginase from *F. proliferatum* in *P. pastoris* was demonstrated in this work, being of great importance for the analysis of new methodologies in search of the production of important drugs in therapy.

## 1. Introduction

l-asparaginase (EC.3.5.1.1; l-asparagine aminohydrolase) catalyzes the deamination of l-asparagine to l-aspartate and ammonia. It has been used as an effective drug in the treatment of acute lymphoblastic leukemia and is industrially produced by the *Escherichia coli* and *Erwinia chrysanthemi*. However, adverse effects such as anaphylactic reactions have been reported in children with leukemia and lymphoma when asparaginase from bacteria was administered [1,2]. In this scenario, it is important to find new sources of l-asparaginase-producing microorganisms that can avoid undesired side effects obtained from bacterial l-asparaginase. The production of therapeutic enzymes in eukaryotic organisms can be a great strategy for reducing adverse effects and for the production of more humanized proteins, with defined glycosylation patterns similar to humans, in addition to having a short growth period, and the absence of endotoxins [3].

Heterologous protein expression as *Komagataella phaffii* (*Pichia pastoris)* strains is already well described in the literature. The yeast *P. pastoris* X-33 was reclassified in *Komagataella phaffii.* These systems are mainly applied in the production of biopharmaceuticals and industrial enzymes [4,5,6,7,8]. The high concept of these yeasts can be attributed to several factors according to Cereghino and Cregg (2000): *P. pastoris* strains have a high capacity for the extra- or intracellular production of heterologous proteins, and molecular techniques for genome modification are also relatively simple and well identified for this microorganism. Yeasts such as *P. pastoris* are capable of carrying out post-translational modifications, such as glycosylation [9]. The use of these yeasts in the expression of heterologous proteins is based on their ability to use methanol as a carbon source. The first step in the conversion of methanol into energy is fundamental in the activity of the enzyme alcohol oxidase (AOX1) found in the peroxisome, converting methanol into formaldehyde [10].

Currently, knowledge about the structure of enzymes is of paramount importance for understanding the mechanisms of catalysis functioning. Understanding conserved amino acid residues in certain regions can establish a direct relationship between the active site and substrate binding [11]. The in silico analysis of protein can reveal important regions associated with drug toxicity or immunogenic reactions [12]. Prior knowledge of the amino acid sequence, as well as the possible three-dimensional structure of the protein, can directly help predict the therapeutic and adverse effects of the treatment.

Therefore, the purpose of this work was to clone and express a filamentous fungal l-asparaginase type II into a *Pichia pastoris* heterologous system, as well as to conduct an in silico analysis of an isolated sequence to preview a 3D structure and its hydrolytic characteristics.

## 2. Materials and Methods

### 2.1. Reagents

l-asparagine, l-proline, trichloroacetic acid (TCA), hydroxylamine hydrochloride, and l-Aspartic acid β-hydroxamate reference standard were purchased from Sigma-Aldrich (St Louis, MO, USA). All the other chemicals used for enzyme assays and characterization were of analytical grade.

### 2.2. Fungal, Bacterial Strains, and Plasmids

The fungus, *Fusarium proliferatum*, used in this study was isolated from the Brazilian Savanna soil that has been previously identified as an l-asparaginase type II producer [13]. The identification was confirmed via ITS, tub2, and ef -1α sequence analysis and deposited in the GenBank under the accession number MT790712 (its), MT815925 (tub2), and MT815926 (ef-1α). The Total Fungal RNA was used as source of genetic material. *E. coli*. TOP 10/P3 (Invitrogen^M^, Carlsbad, CA, USA)—TOPO TA Cloning) was used for pPICZαA vector amplification. *P. pastoris* X-33 (Thermo Fisher Scientific, Waltham, MA, USA) was used as the production of heterologous protein system.

### 2.3. Microorganisms and Growth Conditions

The fungus was maintained at 4 °C and cultured in Petri dishes containing 4% (*w*/*v*) potato dextrose agar (PDA) for 5–7 days at 28 °C. Submerged cultures were carried out at 28 °C on rotary shaker (120 rpm), using modified Czapek-Dox medium (l-proline 1.71%; NaNO3 1.99%; l-asparagine 1.38%; glucose 0.65%; K_2_HPO_4_ 0.0152%; MgSO_4_.7H_2_O 0.052%; KCl 0.052%; ZnSO_4_.7H_2_O 0.001%; FeSO_4_.7H_2_O 0.001%; CuSO_4_.5H_2_O 0.001%, pH adjusted to 6.5 with 5 M KOH).

### 2.4. Sequencing of F. proliferatum l-Asparaginase Gene

Fungal RNA was extracted according to the RNeasy Plant Mini kit method™ (QIAGEN Redwood City, CA, USA). The total RNA concentration was through the absorbance ratio of the final product at 260/280 nm, and the RNA integrity was determined through electrophoresis in a 0.8% agarose gel containing formaldehyde, as described by Sambrook et al. (1989). The RNA obtained was visualized by staining the gel with ethidium bromide at 0.001 μg/mL [14]. The total RNA of *Fusarium proliferatum* was used for subsequent application of reverse transcriptase (Super Script IV Reverse Transcriptase Kit—Invitrogen^TM^) and amplification of the cDNA sequence encoding the enzyme l-asparagine from the constructed primers.

l-Asparaginase biosynthesis gene of *F. proliferatum* was PCR-amplified by using gene-specific primers *FusAsn-For1*: ATGCCCAGCTTTAAACGGCTT; FusAsn-Rev1: GTGCACTCCCGCGTGCTC PCR was performed in 50 µL of reaction mixture, which contained 50 ng of genomic DNA, 0.5 μM of each primer, 200 μM each of dNTP, 1.25 U of DNA polymerase, 1× buffer; 2.5 mM of MgSO_4_, and autoclaved Milli Q water. Amplification was performed with the following conditions: initial denaturation at 94 °C for 3 min, followed by 30 repeated cycles of 94 °C for 30 s, 52 °C for 1 min, 72 °C for 2 min, and final extension at 72 °C for 10 min. PCR amplicons were analyzed on 1.0% agarose gel along with DNA molecular weight marker (GeneRuler 1 kb gene) and documented in a gel documentation system. Agarose gel electrophoresis was used for analysis and quality assessment, DNA quantification, and analysis of DNA fragments.

### 2.5. In Silico Analysis

By using the nucleotide sequence of l-asparaginase, it was possible to compare with other sequences and analyze the three-dimensional structure through a series oftools. SWISS-MODEL workspace, C-I-TASSER Server, Discovery Studio Visualizer (BIOVIA—Dassault Systèmes) v.21.1.0.20298, Modeller (Salilab) v.10.4, were the technological tools used in this work. QMEAN, C-score, and DOPE-Score were used as parameters of quality and reliability. The l-asparaginase sequence were compared with the template PDB 5i3z.1. PDB 5i3z.1 is an *E. chrysanthemi*
l-asparaginase protein with the highest homology found in the databases.

(1) GMQE (Global Model Quality Estimation)—compares the quality of the target-model alignment, wherein the score varies from 0 to 1, reflecting the expected accuracy for the model built through homology. (2) QMEAN—provides an estimate of the geometric quality regarding the positioning and global and individual angulation of each amino acid residue, varying from 0 to −4; whereby, the closer the value is to 0, the better the agreement between the model and structure. (3) C-score—an estimate of the quality of the models predicted using the C-I-TASSER Server. The C-score ranges from −5 to 2, where a high value means high reliability in the model [15,16,17]. The ab initio analysis is thus conducted.

Based an analysis carried out by the Galaxy Web servers, a tetrameric structure can be predicted from templates already deposited in the databases. MolProbity and Ramachandran favored were the quality parameters used in the study.

The nucleotide sequence of the l-asparaginase from *Fusarium proliferatum* was translated into its respective amino acid sequence through ExPASy Bioinformatics Resource Portal. Three-dimensional molecular structures were predicted using C-I-TASSER Server (Contact-guided Iterative Threading Assembly Refinement). Predictions were obtained from the amino acid sequence without the signaling peptide. This server uses multiple deep neural network predictors so that it can identify structural templates from the PDB via multiple threading. This methodology is especially accurate for the sequences that do not have homologous templates in the PDB. A homology search was performed using the linear protein sequence in Uniprot BLAST (https://www.uniprot.org/blast/—UniprotKB accessed on 30 May 2023) with 3D structure database) to evaluate the conservation of the regions relevant to the catalytic activity. The best-matched results were aligned using Jalview software. The Hinge Region (HR) and Active Site Flexible Loop (ASFL), essential regions to the stabilization of the catalytic site [17], were superimposed, and an RMSD analysis was performed (using Discovery Studio Visualizer “Superimpose Proteins” tool), comparing the predicted protein with those found by Uniprot.

### 2.6. Cloning of l-Asparaginase Gene from F. proliferatum

The following primers were used to amplify the l-asparaginase gene into *P. pastoris* X33 strains using PPICZαA vectors by Invitrogen, Carlsbad, CA, USA: Forward primer 5′-ATGCCCAGCTTTAAACGGCTT-3′ and Reverse primer 5-GTGCACTCCCGCGTGCTC-3. Initially, in order to increase the plasmid concentration, *E. coli* strain TOP 10 were transformed aimed to increase the plasmid concentration. Zeocin (25 μg/mL) for 1 day at 37 °C under 200 rpm. was used in the culture medium for selected the transformed strain—(DNA fragments were purified using the Pure Link Quick Plasmid Miniprep kit (Invitrogen, Carlsbad, CA, USA). The DNA fragments were transformed into *P. pastoris* X33 strain (Invitrogen, Carlsbad, CA, USA) via electroporation.

### 2.7. Heterologous l-Asparaginase Expression

The strains stored in a −80 °C freezer were reactivated in 90 mm Petri dishes containing YPD agar medium with 100 μg/mL Zeocin. After 48 h, a single colony was added to a 50 mL falcon tube containing 10 mL of BMGY medium (1% yeast extract; 2% peptone, 100 mM potassium phosphate, pH 6.0, 1.34% YNB, 4 × 10^−5^% biotin, and 1% glycerol). Pre-inoculum cultivation had, as its main objective, the growth of biomass and was maintained at 30 °C, 250 rpm, for 24 h. Growth was monitored by OD600. The pre-inoculum was centrifuged for 10 min at 2000× *g*. The supernatant was discarded, and the pellet was resuspended in 30 mL of BMMY (1% yeast extract; 2% peptone, 100 mM potassium phosphate, pH 6.0, 1.34% YNB, 4 × 10^−5^% biotin and 1% methanol) medium in 250 mL Erlenmeyer’s flasks. Every 12 h, 300 µL of 100% methanol was added to the cultivation process.

The enzymatic expression followed the instructions established in the EasySelect^TM^ Pichia Expression Kit protocol for expression of Recombinant Proteins using pPICZ and pPICZαA in *Pichia pastoris*—Invitrogen^,^ Carlsbad, CA, USA.

### 2.8. Cell Disruption Tests

This work included the insertion of the pPICZαA vector, where, with the use of this vector, there was the expectation of extracellular production of the enzyme of interest, since the α factor of *S. cerevisiae* influences the migration of the protein to the extracellular medium, which, as shown in the results above, did not occur. Thus, an alternative for extracting the intracellular enzyme was the sonication. The intracellularly produced l-asparaginase was extracted according to the methodology described in the literatura, but with modifications [18]. A total volume of 30 mL of suspensions with a concentration of 100 mg of cells/mL was subjected to extraction by an ultrasonic sonicator at a frequency of 40 KHz [18]. Three different methods were tested: *method 1* (30 s ON + 45 s OFF—10 min); *method 2* (1 s ON + 1 s OFF—10 min); and *method 3* (60 s ON + 60 s OFF—10 min). The entire process was carried out in an ice bath to maintain the three-dimensional structure of the protein. At the end of the process, the mixture was subjected to centrifugation for 10 min at 2000× *g*. The assay for quantification of l-asparaginase was applied to the sample supernatant.

### 2.9. l-Asparaginase Activity

Enzyme assay was performed by measuring the l-asparaginase activity in the biomass of extracellular medium through the formation of β-aspartyl hydroxamate from asparagine and hydroxylamine [19]. The reaction mixture contained 100 mM l-asparagine, 1 M hydroxylammonium chloride, 50 mM Tris-HCl buffer (pH 8.6), and diluted enzyme. After 30 min at 37 °C, ferric chloride reagent (10% (*w*/*v*) FeCl_3_, plus 10% (*w*/*v*) trichloroacetic acid in 0.66 M HCl) was added and the samples and centrifuged at 3000× *g* for 5 min at 4 °C. The colored complex was measured at 500 nm. Sample blank was performed with incubation of buffer and sample for 30 min under the same conditions described above, followed by the addition of l-asparagine and hydroxylamine stock solutions after ferric chloride reagent. One unit of asparaginase is defined as the amount of enzyme that formed 1 µmol of β-aspartyl hydroxamate in 1 min.

### 2.10. l-Asparaginase Purification

l-asparaginase was purified on a Hitrap DEAE FF 5 mL ion exchange column (Sigma-Aldrich^®^), followed by the application of 2 mL of sample on the column previously equilibrated with Tris-HCL buffer, pH 8.6, under a flow rate of 0.5 mL/min. For sample application, a column wash was performed using 5 column volumes. After the washing step, sample elution began, varying the NaCl concentrations from 0 to 0.5 M. The tubes were collected in fractions containing 2 mL. Enzyme activity and total protein assays were performed in each tube. The fractions where enzymatic activities were performed were concentrated after lyophilization for the continuation of the characterization tests. SDS-PAGE was performed as described by Laemmli (1970) [20]. Gel was stained with a Coomassie Blue R-350.

### 2.11. Cytotoxic Effect Using MTT Assay

The cytotoxic effect of l-asparaginase on Jurkat cells (obtained from ATCC) was evaluated via MTT [3-(4.5-dimethylthiazol-2-yl)-2,5-diphenyl tetrazolium bromide] assay. For this, 5 × 10^4^ cells were cultured in a 96-well plate with different concentrations of l-asparaginase (0.0005 to 0.01 UI/mL). After 24 h, 10 µL of MTT (5 mg/mL) was added to each well and the plate was incubated for 4 h, protected from light. Then, the plate was centrifuged at 400× *g* × 10 min, and the medium with MTT was discarded, and replaced by DMSO. The formazan crystals were dissolved via homogenization and the optical density was read on a DTX 800 Series multimode detector (Beckman Coulter, Brea, CA, USA) at 570 nm.

## 3. Results

### 3.1. l-Asparaginase Gene Identification

The l-asparaginase gene was amplified using the total cDNA and primers *FusAsn*-For1/FusAsn-Rev1 (Figure 1). The amplification product presented a molecular weight of approximately 1300 bp, indicating the complete amplification of the cDNA of the enzyme. The fragments resulting from the purification were directly sequenced for a previous confirmation. The results of sequencing were compared to the databases in BLAST^®^, which is in accordance with the reported sequence for the cDNA of l-asparaginase from the fungus *F. proliferatum* (NCBI Reference Sequence: XM_031219959.1) [4,7,21,22,23] (Figure 2). The sequence was deposited in the GenBank under the accession number NCBI OQ439985.

### 3.2. In Silico Analysis

After in silico prediction, 10 structures were proposed to correlate the fungal l-asparaginase sequence with the template. The best-scored unit presented a monomeric structure with a molecular weight of 46,444 Da, an isoelectric point of 7.06, and a DOPE-Score of −0.4274. DOPE-score is a statistical parameter to compare structure models via homology.

Appendix A shows the alignment obtained after UniprotKB search in 3D structure database (PDB). There is a low similarity (above 40%) between the Query sequence (*F. proliferatum*) and the already crystallized structures. Since the comparison procedure of conserved domains is used to deduce active sites and the polymeric characteristics, an approach based on the critical regions for the enzymatic activity was performed. Besides the binding site, two relevant areas to the enzymatic activity (Hinge Region—HR—and the Active Site Flexible Loop—ASFL) have already been described as needed for the stabilization of the catalytic site [15,16,17]. Figure 3 presents the alignment of this region and the superimposition of its tridimensional structures. The comparison showed a low RMSD between the predicted structure and all the PDB’s. This result evidences that this vital region is well preserved even in tetrameric ASPases with low similarity.

The active site of l-asparaginases is relatively rigid and conserved as well. The position of the five catalytic residues of interest is relevant to the activity. There are five catalytic residues of interest. Their position is needed for enzymatic activity. There is a threonine in the HR, a tyrosine in the ASFL, and three other residues (a pair of Thr-Asp and Lys) located about 64–67 and 137–140 residues apart, respectively, from the ASFL Tyr. In the *F. proliferatum* ASPase, these residues are Thr13 (HR), Tyr27 (ASFL), Thr92, Asp93 and Lys166 (Figure 3).

The best homology model obtained from C-I-TASSER showed high-confidence results. The C-score was 1.59 C (it has a range of [−5, 2]; a higher C-score implies higher model confidence) and the TM-score was 0.94 ± 0.05 (it has a range of [0, 1], where 1 is representative of a perfect match between two structures) (Appendix A).

From the structure predicted via homology, taking into account the PDB 5i3z.1, a MolProbity Score of 1.61 and a Ramachandran favored of 95.08% can be observed. These values increase from the analysis of the refined structure performed by GalaxyWeb, indicating a greater reliability in the tetramer predicted by this server. Figure 4 shows the Ramachandran plot for the predicted structures.

### 3.3. Heterologous l-Asparaginase Activity

Twenty-one clones were tested for l-asparaginase activity. The highest activity found in cell suspension was expressed in clone #9 (2.84 IU/g), followed by activities in clones 12, 4.1, 13, 8, and 4 (Figure 5). These first experiments were conducted within 72 h of cultivation.

### 3.4. Growth Curve

The growth curve demonstrated an increase in enzyme activity in the clones 4, 4.1, and 8 after 24 h of cultivation and the decay after 48 h. (Figure 6). Clones 9 and 12 showed less activity than shown during screening. After 24 h, the highest activity was observed in strain 8 (3.02 ± 0.05 UI mL^−1^) and later in strain 4 (2.92 ± 0.04 UI mL^−1^). Strain 4 was chosen to continue the cell disruption tests because a greater growth was determined, which could positively alter the extraction process.

### 3.5. Cell Disruption Test

Three different methods were tested in order to observe the best conditions for cell disruption (Figure 7). There was no significant difference between the methods; keeping the cooling conditions similar to those of the active apparatus can be a good strategy for cell disruption. The use of the sonication method may vary with respect to cell wall structure, culture medium, time, and frequency applied [18]. However, some studies show a greater effectiveness of cell disruption when compared to other methods, such as physical and physicochemical methods [24]. In addition, some chemical methods can decrease the yield of the process, denaturing or breaking the protein structure. The processing of the *S. cerevisiae* signal peptide involves a series of steps and can be influenced by several amino acids and by the three-dimensional structure of the enzyme. It is known that the l-asparaginase II in the study is a periplasmic protein and it is possible that there are domains that interact with the cell wall of the yeast *P. pastoris*, preventing the secretion of the enzyme even in the presence of the α factor of *S. cerevisiae* [9,25].

### 3.6. l-Asparaginase Partial Purification

l-asparaginase from *F. proliferatum* produced by *P. pastoris* strain was partially purified in one step as demonstrated in Figure 8. The enzyme activity was found on fractions 6, 7, and 8. The purification factor and yield is shown in Table 1.

The l-asparaginase treatment induces Jurkat cell death.

To evaluate the cytotoxic potential of l-asparaginase derived from *Fusarium proliferatum* in Jurkat cells, we performed the MTT assay. For comparative purposes, we also evaluated the cytotoxic effect of standard asparaginase obtained from *Escherichia coli* (Sigma-Aldrich). Importantly, while standard l-asparaginase did not impair Jurkat cell viability at the investigated doses, the treatment with *Fusarium proliferatum*-derived l-asparaginase was able to induce Jurkat cell death at concentrations of 0.005 (*p* = 0.04), 0.007 (*p* = 0.009), and 0.01 UI/mL (*p* = 0.001) (Figure 9).

This section may be divided by subheadings. It should provide a concise and precise description of the experimental results, their interpretation, as well as the experimental conclusions that can be drawn.

## 4. Discussion

A study carried out by Dwivedi in 2014 compared several amino acid sequences deposited in the NCBI database of l-asparaginase protein from different sources: bacterial, fungal, and plant sources. A similarity can be observed in the sequences from fungi and bacteria, where about 10.77% would be represented by the amino acid alanine, which suggests an important role for this amino acid in the composition of l-asparaginase. In addition, the study demonstrated two glycine residues in conversation in all analyzed sequences, which represents an important function for this residue in the evolution of sequences of l-asparaginases of prokaryotic origin [26].

The model obtained is a monomer, and all results are comparable to monomers of tetrameric ASPases. Thus, it is most probable that the *Fusarium proliferatum* asparaginase is also a tetramer.

It is noteworthy that the coding sequence of the l-asparaginase enzyme used in this work has 437 amino acid residues and that the structural modeling calculations are based on structures already available; and currently, there is no crystallized fungal l-asparaginase structure for comparison. Thus, there is a need to purify this enzyme and perform X-ray crystallography for a better understanding of the three-dimensional structure, as well as its kinetic and dynamic parameters.

Lima et al. (2020) cloned and purified a l-asparaginase from *E. coli* in *P. pastoris,* obtaining an activity of 2.98 U/mg of protein after 48 h [27]. Likewise, extracellular ANSase recombinant in *P. pastoris* was purified, expressing an activity of about 2.81 IU/mL in a work performed by Sajitha et al. (2015) [28]. Based on these works, we produced 245 UI in 1 L of cultivation with approximately 86 g of cell.

Freitas et al. expressed a *Penicillium*
l-asparaginase in a *Pichia pastoris* expression system; this work was the first to present the expression of a filamentous fungus enzyme in a yeast expression system [4]. The results shown in the cloning of the *F. proliferatum* sequence are similar to those demonstrated in the previous work. It is worth noting that despite the signal peptide sequence to produce the extracellular enzyme, both studies demonstrated intracellular production, emphasizing the enzymatic function already described in the literature.

In a systematic review on the production of l-asparaginase published by Souza et al., it was demonstrated that the time required for fungal cultivation in submerged fermentation was, on average, 4 days. From the data obtained in the growth curve above, it can be seen that 24 h was necessary for the maximum production of l-asparaginase from *F. proliferatum* in *P. pastoris* [29].

Despite the low yield obtained through the sonication method, as already described by other authors, the presence of the enzyme in the extracellular medium is important for a better continuation of the purification, characterization, and cytotoxicity steps.

Thangavelu (2022) sonicated a *Chatemonium* sp. sample containing 2% *w*/*v* of cells. Ultrasonication was performed with 30 s ON and 30 s OFF for 10 min with intermittent cooling, close to the values used in method 1 of this work. After the purification steps, the enzyme shows a specific activity of 1175 U/mg [30]. *P. pastoris* samples that were previously sonicated were at a concentration of 10% (*w/v*), which may have reduced the efficiency of the sonication process. Methods seeking to optimize the enzymatic extraction process need to be explored with a view to increase scale and industrial production. However, the objective of this work was to explore and identify the production of a fungal l-asparaginase in a heterologous expression system, and despite the low activity, it was concluded that it was possible to produce the l-asparaginase from *Fusarium proliferatum* intracellularly, which has been proven through the cell disruption tests.

In previous studies, carried out by Freitas et al. (2021), the production of l-asparaginase from *Fusarium proliferatum*, after the cell disruption process was around 1.86 ± 0.12 U mL^−1^. In this work, the heterologous production of the same enzyme as well as the yield were higher than in the previous study, demonstrating the importance of exploring new strategies for the production and cultivation of microorganisms [13]. Systems using *P. pastoris* have the advantage of methanol induction, a variable that can be optimized under cultivation conditions. In addition, other conditions such as pre-inoculation time, glycerol concentration in the BMGY medium, and changes in methanol concentration during the process can be explored in order to reach an optimal condition.

It is possible to observe the presence of two major bands in the electrophoresis gel: the first band being close to 45 kDa would be suggestive of the already mentioned l-asparaginase monomer with 46.4 kDa, and the second band with approximately 30 kDa. Other purification steps could be used in order to completely purify the enzyme; however, upon observing a low yield of 2.95%, it was decided not to use another step. Importantly, the heterologous ASNase produced in *P. pastoris* was partially purified with a single chromatographic method. The decrease in the number of downstream steps, as well as the fast cultivation and good yield in the upstream processes, reduce the enzyme production cost and can help new technologies to obtain and improve new products [31].

In a study published by Kleingesinds et al. (2023), it was possible to obtain a yield of approximately 55% and a Pf of 70.9 after four steps of the purification process, activated in an extracellular enzyme with a high degree of purity [31]. It is known that extracellularly produced enzymes generate a greater possibility of obtaining high yields. In the current work, the ASNase from the *F. proliferatum* confirmed intracellularly, which changes the decisions for different separation and purification steps as well as in decreasing order, reach the yield and the Pf.

Arumugam and Thangavelu (2022) required the production and purification of an intracellular ASNase from *Chaetomium* sp., where the results of the purification factor (Pf = 2.45), using two steps to purify the enzyme, were similar to those presented by this work [29].

There are no reports in the literature of l-asparaginase from filamentous fungi produced and purified in *P. pastoris*. This is the first work showing purification results of this enzyme in this expression system. It is important to point out that the production steps as well as the variables involved in the process were not optimized, thus concluding that the results of this work can be even more promising when these aspects are explored.

Yap et al. reported that the concentration of 0.67 IU/mL of unpurified *F. proliferatum* l-asparaginase was sufficient to eliminate 50% of viable Jurkat cells within 24 h of treatment [32]. These results corroborate that the ASNase produced by this fungus can have a favorable use in pharmaceutical and food industries. Some parameters have yet to be evaluated, but it is known that enzymes produced by eukaryotic organisms are less immunogenic due to post-translational modifications. In a work published by Arumugam and Thangavelu, it took 72 h to observe the decrease in the cell viability of the MOLT-4-lineage cells at concentrations greater than 13.04 µM [29].

## 5. Conclusions

This work has so far presented results of the expression and quantification of l-asparaginase in expression vectors of *Pichia pastoris* X33 transformed with the sequence of the enzyme of fungal origin from *Fusarium proliferatum*. Despite the presence of *Saccharomyces* α factor in the pPICZαA vectors to stimulate the migration of the enzyme to the extracellular medium, an intracellular production of the enzyme was noted, as shown in the results of the quantification of the enzyme in biomass. The method of cell blocking with an ultrasonic sonicator tip proved to be effective in transmitting the enzyme of interest, increasing the enzyme amount by about 8 times.

Twenty one clones were tested. Clones 4 and 8 were selected for the continuation of the work, since they have a relevant enzymatic and biomass production within 48 h of cultivation using a concentration of 1% methanol every 12 h.

However, there is still a difficulty in carrying out structural homology modeling since there are no structures of fungal l-asparaginases deposited in protein databases. Ab initio modeling proved to be effective in the structural predictability of the enzyme. In silico analysis was able to predict thermodynamically stable monomer structures. The occurrence of conflicts in certain regions of the protein regarding the positioning of certain amino acid residues is inherent to the process. Comparison with crystallized structures of *E. coli* facilitated the construction process of tertiary structures. Despite that, for a better accuracy of the process, comparison with structures of eukaryotic l-asparaginases is necessary.

## Figures and Tables

**Figure 1 pharmaceutics-15-02352-f001:**
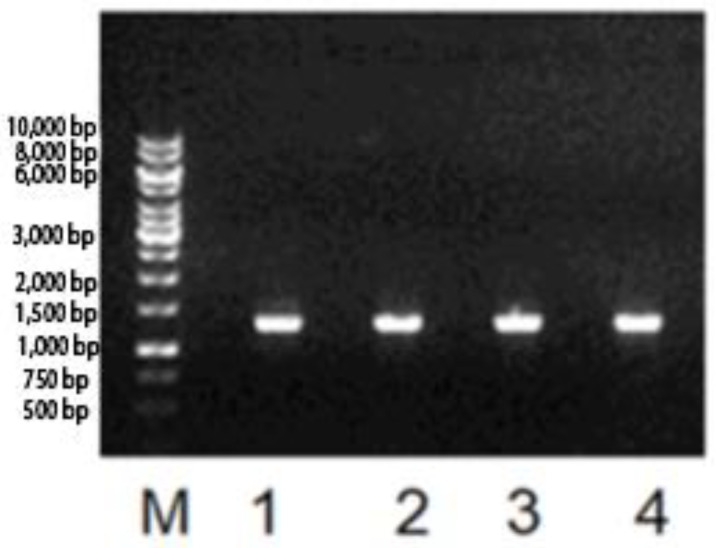
Electrophoretic profile on 1% agarose gel in 1× TAE buffer for fragments amplified with oligonucleotide primers (primer). From left to right M refers to the 1 kb “ladder” DNA. Lanes 1, 2, 3, and the band corresponding to 1300 bp amplified with the primer pair FusAsn-For1/FusAsn-Rev1.

**Figure 2 pharmaceutics-15-02352-f002:**
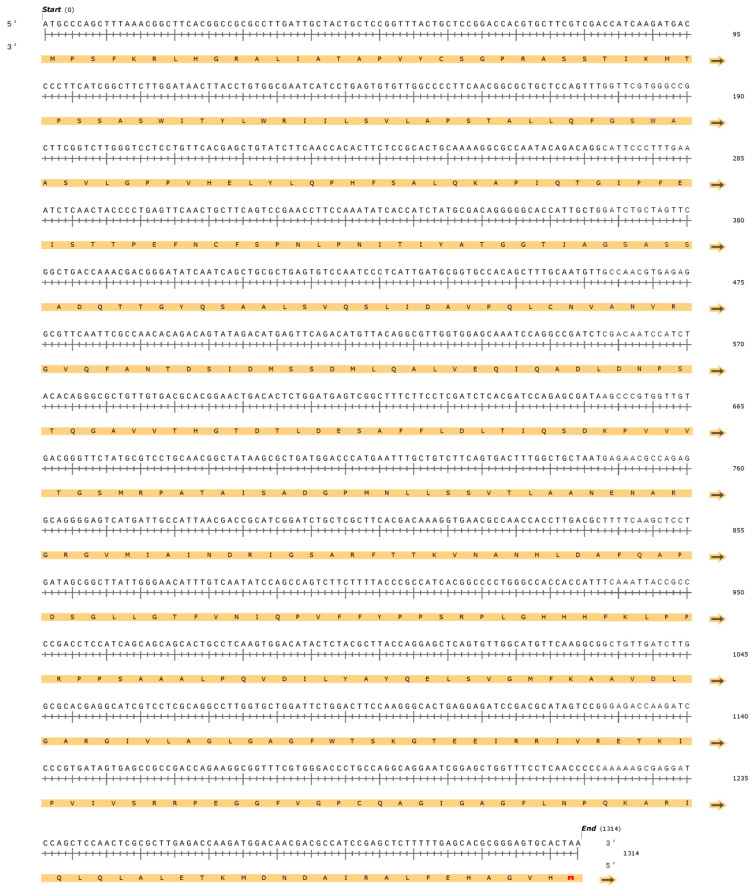
*F. proliferatum* l-asparaginase sequence and possible encoded amino acids. ATG start codon represents methionine and TAA stop codon.

**Figure 3 pharmaceutics-15-02352-f003:**
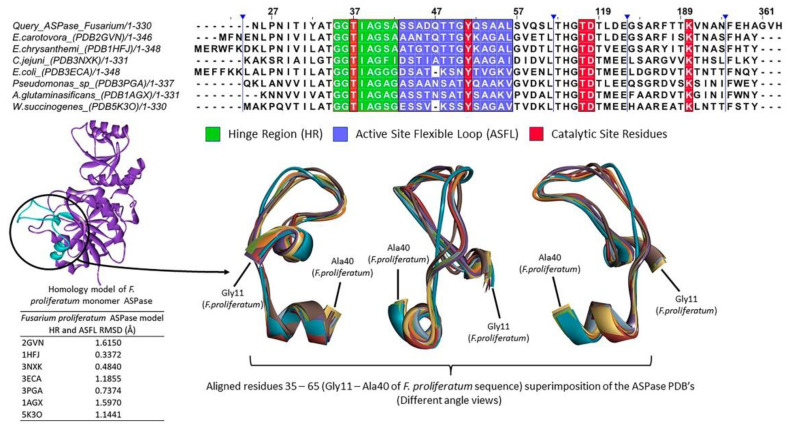
Structural analysis of conserved residues of the hinge region, active site, and catalytic site.

**Figure 4 pharmaceutics-15-02352-f004:**
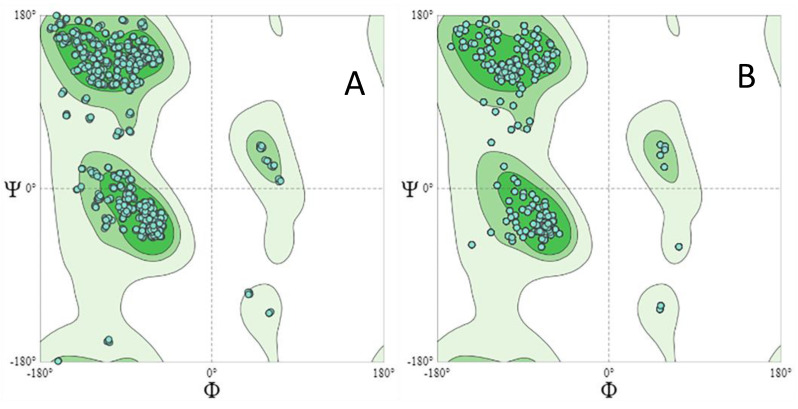
(**A**). Ramachandran plot for the structure created through homology using the PDB 5i3z.1 template. MolProbity Score: 1.61; Ramachandran favored: 95.08%. (**B**). Ramachandran plot for the structure created through analysis performed by GalaxyWeb. MolProbity Score: 2084; Ramachandran favored: 96.7%.

**Figure 5 pharmaceutics-15-02352-f005:**
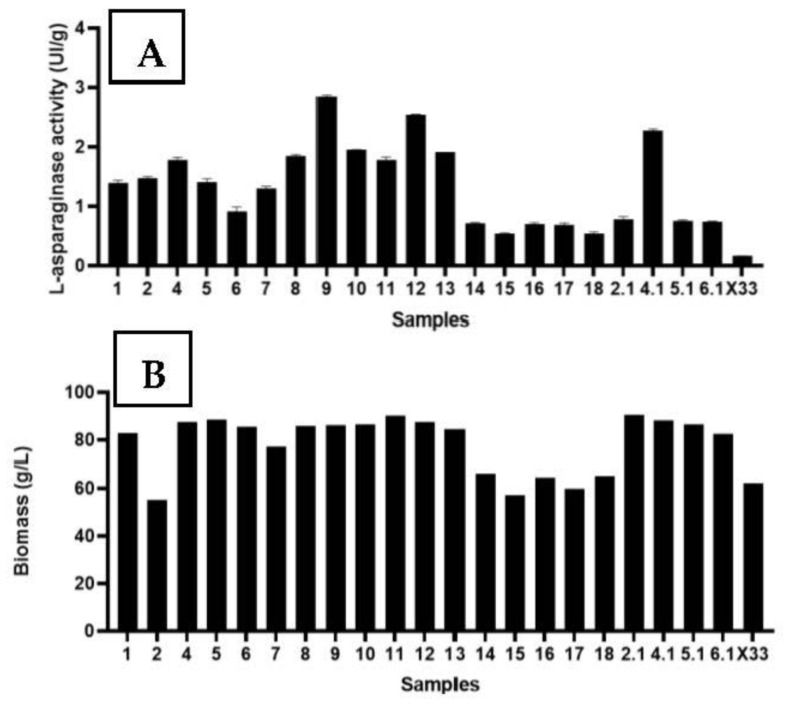
(**A**)—l-asparaginase activity in the 21 tested clones. (**B**)—Biomass growth (g/L). Strain X33 is considered as negative control—the absence of PPICZ vector and *F. proliferatum*
l-asparaginase sequence.

**Figure 6 pharmaceutics-15-02352-f006:**
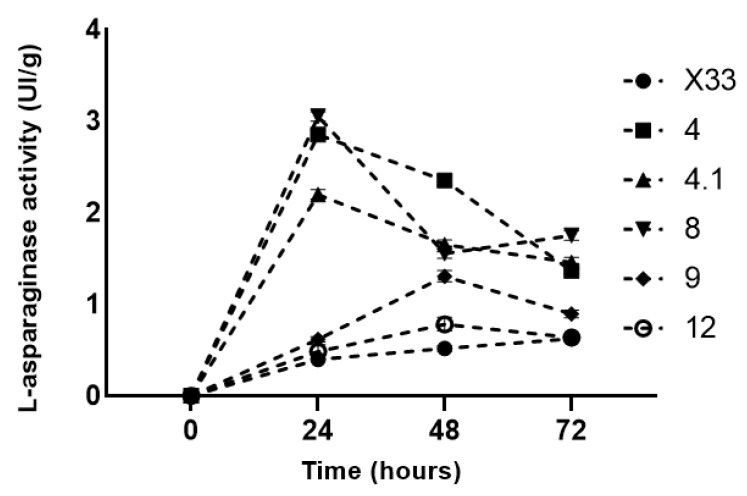
Growth curve for the analysis of the best cultivation condition in relation to the time variable. Clones 4, 4.1, 8, 9, and 12. X33 strain as negative control. Assays were performed in triplicate. Results are presented as mean and standard deviation.

**Figure 7 pharmaceutics-15-02352-f007:**
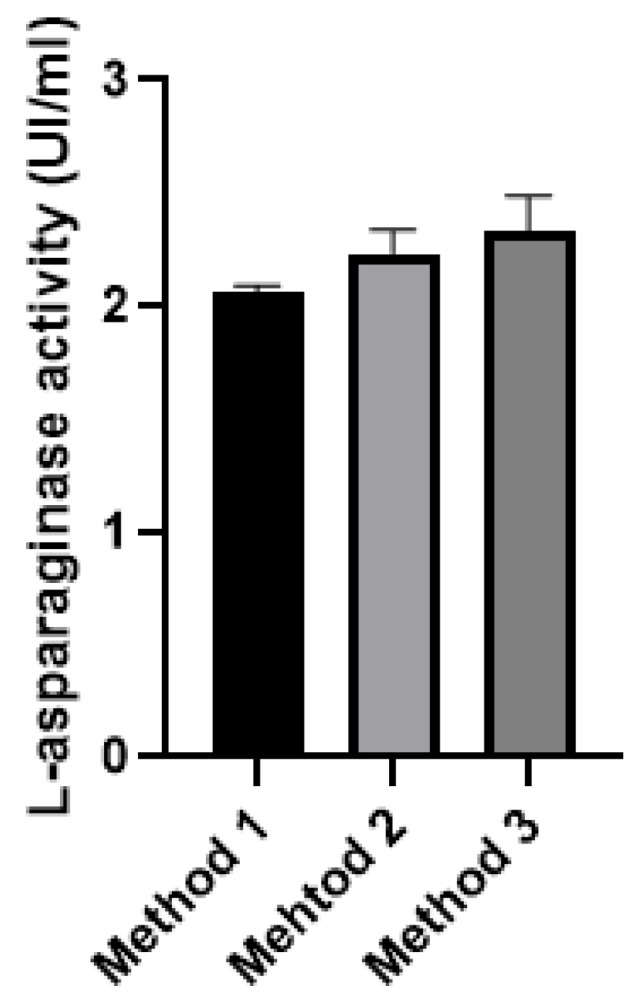
Results for sonication methods. Method 1 (30 s ON + 45 s OFF—10 min); Method 2 (1 s ON + 1 s OFF—10 min); and Method 3 (60 s ON + 60 s OFF—10 min). Assays were performed in triplicate. Results are presented as mean and standard deviation.

**Figure 8 pharmaceutics-15-02352-f008:**
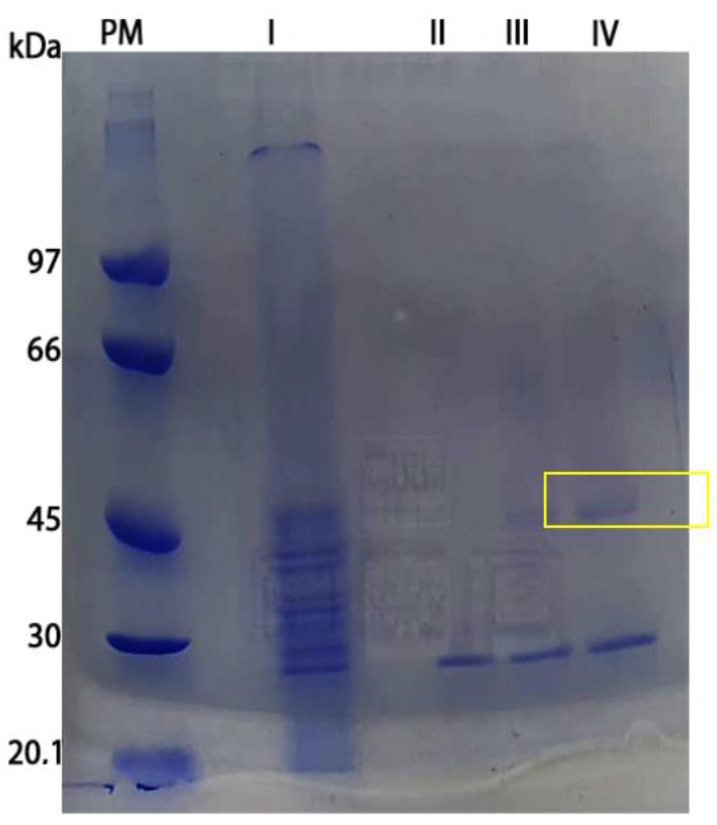
SDS-PAGE of l-asparaginase purification from the DEAE column. PM—protein marker (Calibration Kit Low Molecular Weight For Electrophoresis—Cytivia 17-0446-01); Lane I—crude extract; Lane II—fraction 6; Lane III—fraction 7; Lane IV—fraction 8. Yellow marker—possible L-asparaginase monomer.

**Figure 9 pharmaceutics-15-02352-f009:**
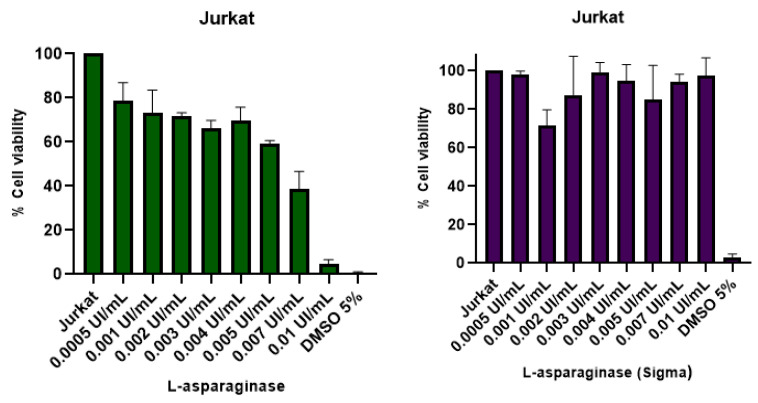
Jurkat cells’ viability against different concentrations of l-asparaginase form *F. proliferatum* compared with the *E. coli* standard l-asparaginase (Sigma Aldrich). Assays were performed in triplicate. Results are presented as mean and standard deviation.

**Table 1 pharmaceutics-15-02352-t001:** Purification steps of *F. proliferatum* l-asparaginase expressed in *P. pastoris*.

Purification Step	Volume (mL)	Total Activity (U/mL)	Total Protein (mg)	Specific Activity (U/mg)	Purification Fold	Yield (%)
Crude Extract	25	2.3	2.2	1.04	1.00	100
DEAE FF	25	0.09	0.29	0.31	0.29	7.25
Concentrated Enzyme (10×)	2.5	1.16	0.94	1.23	1.18	2.95

## Data Availability

The data presented in this study are available in this article (and Appendix A).

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
