# Peer review of "l-Asparaginase Type II from Fusarium proliferatum: Heterologous Expression and In Silico Analysis"

_pharmaceutics, 2023, doi:10.3390/pharmaceutics15092352_

Round 1

Reviewer 1 Report

The manuscript “pharmaceutics-2558328” is recommended for accept after major revision”.

 This manuscript covered the cloning and expression of a L-asparaginase from Fusarium proliferatum in Pichia pastoris, as well as its in silico analysis. Considering the importance of  L-asparaginase, It is believed that the results would be highly valuable for the readers in search of the production of important drugs in therapy. 

Major comment: In table 1, the specific activity of crude extract was 1.04 U/mg. Then, the specific activity was decreased to 0.31 U/mg. Since the specific activity reflected the purity, which contradicted the information in Figure 8. It is also required to explain why the specific activity was increased up to 1.23 U/mg. 

Minor comment: the quality of the manuscript should be further improved. Please thoroughly check the whole text.

Line 217 to 225, add the space between number and it unit.

In Figure 1, the unit for each band in the DNA ladder should be bp.

In Figure 6, 7 and 9, indicated how many tests was run for the calculation of standard deviations.

In Figure 8, Fraction at the end of figure legend should be deleted.

The English Language should be further improved.

Author Response

 This manuscript covered the cloning and expression of a L-asparaginase from Fusarium proliferatum in Pichia pastoris, as well as its in silico analysis. Considering the importance of  L-asparaginase, It is believed that the results would be highly valuable for the readers in search of the production of important drugs in therapy. 

Major comment: In table 1, the specific activity of crude extract was 1.04 U/mg. Then, the specific activity was decreased to 0.31 U/mg. Since the specific activity reflected the purity, which contradicted the information in Figure 8. It is also required to explain why the specific activity was increased up to 1.23 U/mg. 

               The initial activity was analyzed in the fraction from the purification process. Since the sample undergoes a dilution process at this stage, it was necessary to lyophilize the active fraction for a better and real quantification of the enzymatic activity. The correct specific activity being 1.23 U/mg.

Minor comment: the quality of the manuscript should be further improved. Please thoroughly check the whole text.

Line 217 to 225, add the space between number and it unit. - DONE

In Figure 1, the unit for each band in the DNA ladder should be “bp”. - DONE

In Figure 6, 7 and 9, indicated how many tests was run for the calculation of standard deviations. - DONE

In Figure 8, “Fraction” at the end of figure legend should be deleted.  - DONE

Changes requested by reviewers are marked in yellow.

Reviewer 2 Report

The work describes for the first time the partial purification of L-asparaginase from filamentous fungi overexpressed in P. pastoris. The method described could help purify other types of enzyme from fungi. The work is therefore interesting, but there are some points that need to be clarified.

In figure 8, the authors should circle on the gel where the asparaginase is located.

A mass spectrometry study would be interesting to try identifying the contaminating protein at about 30kda. Isn't this a hindrance to the rest of the studies? The authors should comment.

In addition, the authors mention the existence of tetrameric ASPases. A comment about the oligomeric state of ASPase should be added in the introduction, what is known in literature. In addition, there are algorithms that predict the oligomerization state of proteins. it could be interesting to apply them (e.g. the webserver Galaxy) and add the results in the in silico analysis.

In the materials et methods, several programs, SWISS-MODEL, ITASSER, Discovery Studio Client, Modeler 3D and Profiles, are cited as having been used to build the enzyme's 3D structure. A comparison of the results obtained for each program should be made.

A ramachandran plot would be useful to show to support the model.

The author mention once in the text the « L-asparaginase II », is there a L-asparaginase I ?

The authors should specify in the introduction which enzyme they are working on.

Minor points :

- For the in silico analysis, the authors should clearly mention what are the nature of the different enzyme templates. Are they all L-asparaginase ?

- What's the point of figure 4 compared with figure 3 (with the homology model of the enzyme)? Figure 4 seems unnecessary.

Minor editing of English language required

Author Response

The work describes for the first time the partial purification of L-asparaginase from filamentous fungi overexpressed in P. pastoris. The method described could help purify other types of enzyme from fungi. The work is therefore interesting, but there are some points that need to be clarified.

In figure 8, the authors should circle on the gel where the asparaginase is located. - DONE

A mass spectrometry study would be interesting to try identifying the contaminating protein at about 30kda. Isn't this a hindrance to the rest of the studies? The authors should comment.

 The search for a protein that presents 100% purity is necessary to carry out several other studies. However, it is known that clinically, commercial L-asparaginase also has impurities. For structure determination by x-ray crystallography it is necessary to obtain a single band and a high degree of purity. The applied purification method was not able to separate the two proteins, however, the application of one more method could further dilute the sample resulting in a lower yield.

 In addition, the authors mention the existence of tetrameric ASPases. A comment about the oligomeric state of ASPase should be added in the introduction, what is known in literature. In addition, there are algorithms that predict the oligomerization state of proteins. it could be interesting to apply them (e.g. the webserver Galaxy) and add the results in the in silico analysis. - DONE

In the materials et methods, several programs, SWISS-MODEL, ITASSER, Discovery Studio Client, Modeler 3D and Profiles, are cited as having been used to build the enzyme's 3D structure. A comparison of the results obtained for each program should be made.

 A ramachandran plot would be useful to show to support the model. – DONE Figure 4.

 The author mention once in the text the « L-asparaginase II », is there a L-asparaginase I ?

The authors should specify in the introduction which enzyme they are working on. DONE

Minor points :

- For the in silico analysis, the authors should clearly mention what are the nature of the different enzyme templates. Are they all L-asparaginase ?

For homology analysis, the template used was PDB 5i3z.1, derived from E. chrysantemi Asparaginase. For the ab initio analysis of the monomeric structure, several structures already deposited in databases were used, most of which were L-asparaginase. The other structures used are from proteins that have a higher percentage of homology (Supplementary Material)

 - What's the point of figure 4 compared with figure 3 (with the homology model of the enzyme)? Figure 4 seems unnecessary. - DONE

Figure 3 shows the overlapping of certain regions discussed in relation to the L-asparaginase coding sequence produced in this study. Figure 4 shows a possible monomeric three-dimensional structure produced through the ITASSER servers by ab initio analysis.

Round 2

Reviewer 1 Report

Accept in current form.

The quality of English language is fine.

Reviewer 2 Report

The authors have responded to the majority of comments/suggestions.